

# [ITS 2019 Ai Tutorial]

# ITS 2019 Artificial Intelligence Code Tutorial

**A small tutorial session featuring a live Ai session at UWI MONA for [ITS 2019](), where a basic [Artificial Neural Network]() will be written from scratch & discussed, led by Jordan Micah Bennett**

**ITS Website: [https://its2019.iis-international.org/](https://its2019.iis-international.org/)**

# Contents

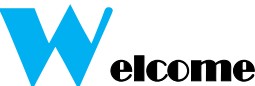

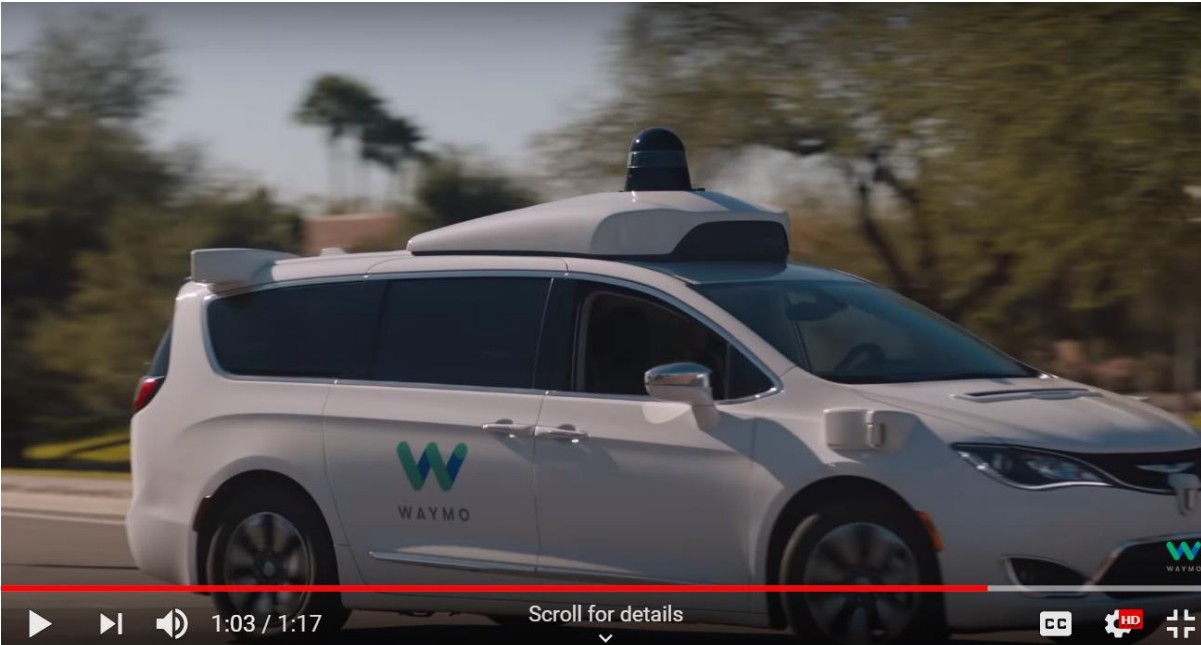

Figure 1: *A self-driving car driving without driver* in the State of Phoenix. (This self-driving taxi service, which is powered by artificial intelligence, *was made available to the public on December 5, 2018*!)

...

Brain inspired computer code or smart apps, called **AGI or Artificial General Intelligence** (predicted to happen by as soon as 2029 or sooner), will perhaps one day be mankind's last invention! (AGI research is beyond the scope of this tutorial, and interested parties can see *MIT's AGI course here.)*

For now though, AGI's predecessor, called **Artificial Narrow Intelligence**, also called **Artificial Intelligence**, can do amazing stuff like diagnose diseases better than human doctors, enable self driving cars, or give game characters the ability to learn to navigate game environments without human aid!

Crucially, *where for example, Ai is already enhancing banking*, fortunately the Jamaican government has recognized the impact that artificial intelligence already brings, and what shall likely happen futureward. I speak more about this in the gleaner articles found on the newspaper tab on this experimental platform of mine.

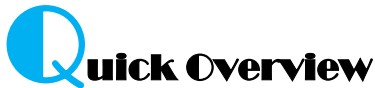

# Quick Overview

1. All of successful artificial intelligence algorithms today perform something called error minimization.

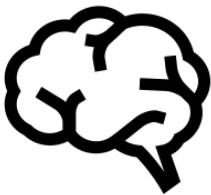

2. They work *similar* to how biological brains work.

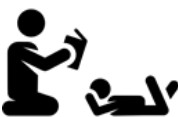

3. For eg, with a *high error rate*, a child will first *wrongly* identify objects in the world in his/her earlier years of life.

3.b. That error rate gets *smaller* or is minimized, as the child gets better at identifying objects in the world; in the early years, a parent can guide the child by saying this is a cat or this is a dog etc, i.e. the parent helps the child to *correctly label* objects in world.

3.c. After a while, even without parental guidance, the child will be quite good at identifying objects, and his/her error rate at object identification would have been *minimized substantially*.

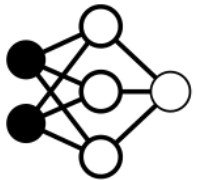

4. Artificial Intelligence, such as the artificial neural networks work in a similar way; for a particular task, they start out *terrible* with high error rates, then they get far better after being exposed to many instances of correctly labelled things, until they get to a point of doing the task well, even without being exposed to correctly labelled data.

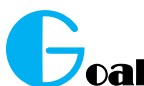

# Goal

1.  The goal is to aim to equip Jamaica with more programmers that are willing to use learning algorithms, to solve tasks that are hard for human programmers to approximate. There's a ==common misconception== that these smart models require Einsteinian intellect to approach, which this tutorial will seek to dispel, after which the programmer would have successfully written/somewhat understood a smart model.

2.  ==Why?== There are just some tasks that no human programmer nor set of programmers have been demonstrated to have sat down in order to explicitly write software codes that anticipate large numbers of scenarios. These types of non-human predictable tasks are getting more numerous as data like banking data grows.

3.  Examples of some tasks with human unpredictable scenarios include image detection pertaining to self-driving cars, automated disease diagnosis of humans or plants, etc.

4.  Artificial neural networks are one of the most ==popular== learning algorithms, ==and in June, we'll try to spend 45 minutes to write a basic artificial neural network in java from scratch without using libraries. (The aim is to guide interested participants along in writing a basic artificial neural network, as I write said code.)==

    This exercise is really geared towards preparing the programmer to better apply machine learning libraries.

5.  Though optional, understanding basic neural nets (even a non-math heavy, but programmatic understanding instead) can afford the programmer better grasp of applying machine learning libraries such as *tensorflow* built by Google or *azure ml* by Microsoft or other ones by other parties like the one I used to do the smart malicious weed detection algorithm. (We ==*won't*== go through the math behind basic neural nets, but we'll go through a relatively simple live programming example instead.)

1.  Our model will do xor prediction, which is a sort of "hello world" neural network example. The same principles largely apply in more complicated neural nets, found for eg, in self-driving cars.

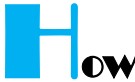

a. Given some inputs, neurons aka nodes do a bunch of calculations in order to represent the input wrt some correct label, then return an output which will be close to the correct label.

b. We can talk about how this works in terms of loops. Three types of loops get this done;

- Loops that live in each ***neuron***, that enable each neuron to do calculations with respect to other neurons.
- Loops that create and organize the neurons above into layers, i.e. our ***neural network.***
- Loops responsible for generating a training scenario that feeds multiple sets of input training data to the neural network, i.e. our ***mainExecutionClass***.

c. In each training scenario, neurons take a set of inputs, do calculations based on those inputs wrt the correct label, and then produce guesses that are initially terrible. So the neural network is being "supervised" in terms of the correct labels.

- {Eg 1 Training cycle : take input set [1,0] wrt expected correct label = 1, generate guess = .18, which is terrible}
- In other words, the neural network takes many input/correct label pairs, and returns a guess aka answer each cycle, where input=[1,0] and the label =1, is an example of an input/label pair, and the guess will be a real number like .18.

d. After roughly 800 training cycles in the case of our xor neural network model, aka taking 800 sets of inputs/correct label pairs, the model would have returned sensible guesses that would be close to the correct labels.

# 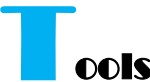ools

Required tools participants are to bring to this event:

- Laptop

- Familiarity with any language that supports classes and lists. (Preferably Java experience, and a Copy of BlueJ IDE, although I will have a copy of BlueJ for those who desire it)

- 45 minutes to an hour of time to spare, on the day of the tutorial, to construct an artificial neural network like this prior example written by Jordan.

- Willingness to prepare oneself for increased automation of tasks/jobs, while eventually being able to leverage Ai to solve seemingly intractable tasks.

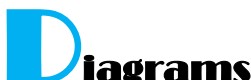

## Diagram 1: Neural network learning structure

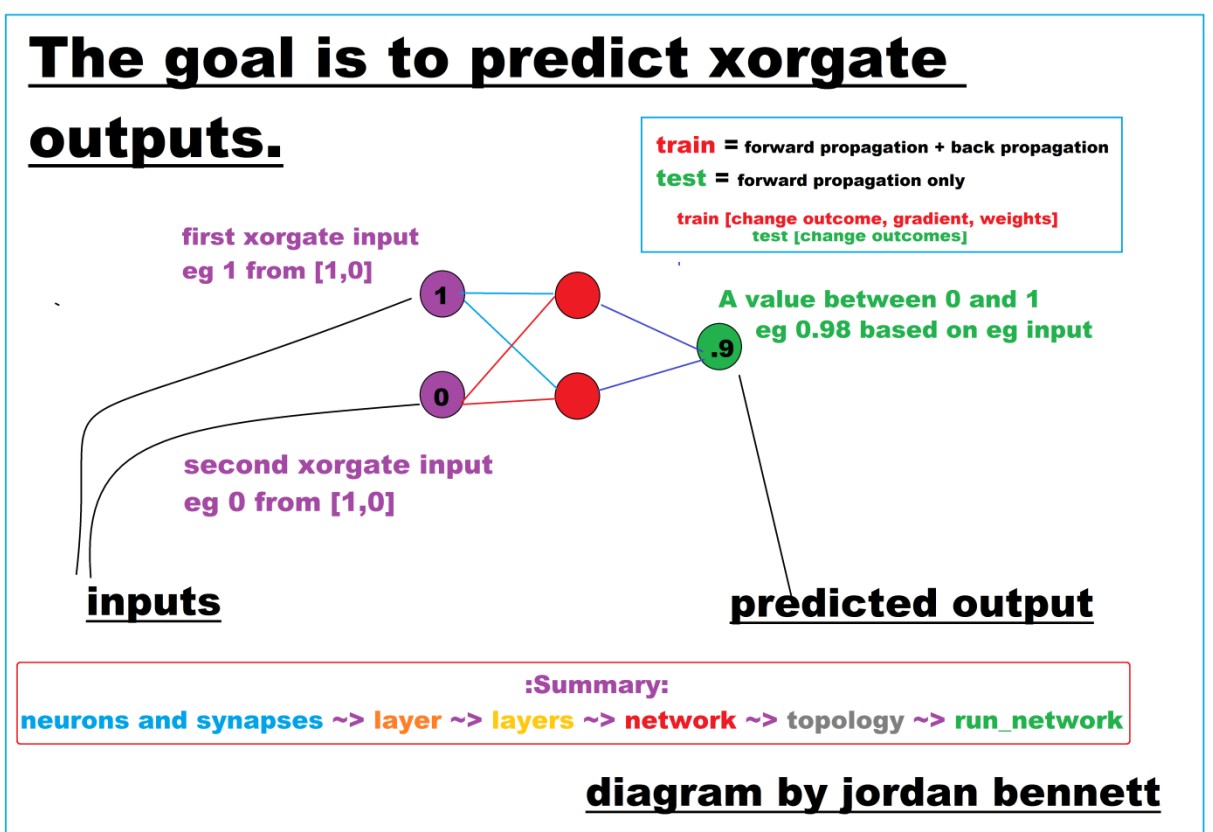

## Diagram 2: xor gate table

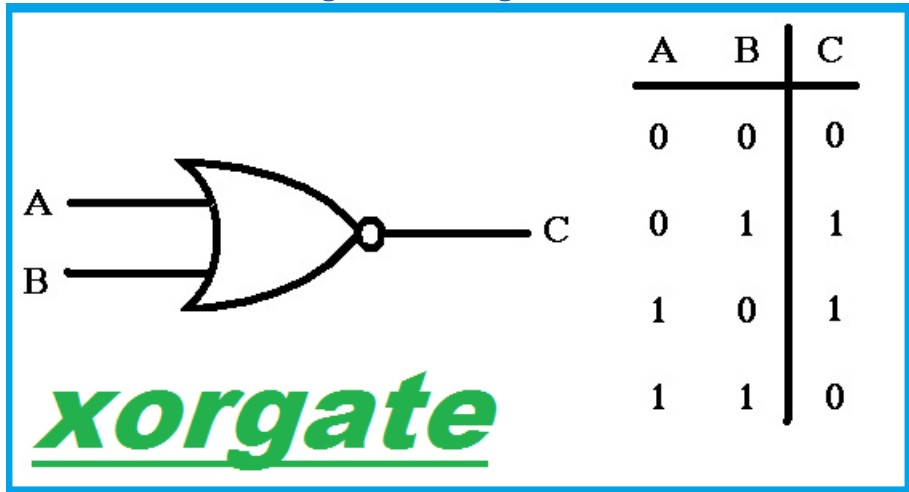

Regards,

Jordan Micah Bennett, author of the weekly Jamaican-newspaper Gleaner column: "Artificial Intelligence and the economy", inventor of the "Supersymmetric Artificial Neural Network" and author of "Artificial Neural Networks for kids". Jordan is also working hard, trying to build something called the MLJI.

