# OpenReview forum: "Artificial Neural Network Code Tutorial "
_ECMLPKDD.org/2020/Workshop/TeachML — Submitted to ECML PKDD 2020 TeachML_

### Official Review · AnonReviewer2 · 2020-07-17
**A valuable idea and contribution but not in a suitable format for this workshop**

**Rating:** 4
**Confidence:** 5

**Review:**

The submission presents an announcement that was used to advertise a workshop on programming artificial neural networks at ITS 2019 in Jamaica. The tutorial's goal was to show how to program a simple neural network that is able to decide the XOR function in a 45 minutes live coding session. The tutorial also offered some source code accompanying the material: https://github.com/JordanMicahBennett/BASIC-ARTIFICIAL-NEURAL-NETWORK_FROM-LIVE-JAVA-SESSION

While I do think that the tutorial is valuable (esp. for the Jamaican programming/growing AI community), I think the submission is not suited well for this particular workshop: 1. it does not adhere to the formatting guidelines stating the ICML template, 2. it is not properly anonymized. I there recommend to reject this submission.

It would have been a great opportunity to share some of the lessons learned from the tutorial held in 2019. I very much encourage the author to resubmit such an experience report to the next edition of the workshop. Still, attending the workshop and contributing to a shared position paper could be envisioned.

---

### Official Review · AnonReviewer3 · 2020-07-17
**Contribution that shows high motivation to teach ML, but does not comply with submission format requirements**

**Rating:** 4
**Confidence:** 5

**Review:**

The submission consists of an announcement concerning a live coding tutorial
given in 2019 at the 15th International Conference On Intelligent Tutoring
Systems (ITS2019). It shows the author's enthusiasm for the covered
topic and for teaching the latter.

However, the submission fails to comply with this workshop's submission format,
which is the ICML 2019 LaTeX template. Additionally, the authorship was not
properly anonymized. Therefore I cannot recommend this submission to be
accepted, which is unfortunate, given the author's clearly high motivation.

Nevertheless, I strongly suggest to submit a revised version to a later
iteration of this or another workshop with a similar scope. In order to make
future submissions more successful, I'd like to point out some aspects that may
be improved.

Even though this workshop's call for papers states that submissions shall
"present *or* discuss a teaching activity related to machine learning", a
discussion of the experiences from teaching at the ITS2019 workshop would have
greatly increased the value of this submission. Additionally, the submission
is, as it seems, an unaltered copy of a workshop announcement for ITS2019,
rather than an article targeted to the present workshop. In terms of format,
future submission might benefit from (i) usage of the common abbreviation "AI"
(uppercase) rather than "Ai", (ii) writing out other acronyms such as ITS or
UWI that might not be known by all readers, (iii) reduced usage of everyday
language (optional, depending on personal style as well as the submission's
target audience).

The submission's content has potential and can be a valuable contribution,
which I hope to see in future workshops.

---

### Official Review · AnonReviewer1 · 2020-07-23
**Live Coding an XOR Predictor**

**Rating:** 3
**Confidence:** 5

**Review:**

I appreciate the efforts by the authors to submit an article to ECML2020's teaching ML workshop. The submitted document does not comply with the layouting requirements our workshop has. Further, the article exceeds the page limit by more than a factor of 2. Both of these aspects make it unlikely to be accepted. A majority of the text is written in colloquial language which to me does not aspire to the neutral character an article needs to be written in. Further, the author gives away his name which makes a double blind review process impossible.

To be precise:

- page 4: yields a general praise of today's Machine Learning (ML) given it's applications which is fine, but in the wrong place to illustrate a teaching approach for ML

- page 5: contains appealing metaphors to compare the inner workings of a Machine Learning algorithm, but fails to connect these insights to teaching approach that is to be presented

- page 6: contains again a praise of Machine Learning in general; point 5 tries to motivate the teaching demo by aspiring to prepare programmers for the use of tensorflow or other tools (which are vaguely touched)

- page 6: the bottom of the page lists a point one after point 6 which is most likely a typo

- page 7: describes the live coding demo in a mixture of bullet points and prose text; here the author describes how a predictor of an XOR operation is live coded during the demo. In itself, this is a wonderful idea as it is simple enough to construct a hard coded network around this. Unfortunately, the way the paper is written hinders a clear understanding on how the demo is provided and what the author wants to achieve his listeners .

To improve the paper, I suggest the following improvements:

- follow the layout requirements of our workshop
- keep the volume within 4 pages (without references)
- present the learning goals of the live demo
- present 2-3 learner profiles of the people you expect to participate in the live demo
- describe key aspects of the code to be demonstrated
- show how the presented code ties back to the motivation, the learner profiles and learning goals
- present evaluation data of how your learners liked the demo and on what they learned

Given the outstanding ideas hidden in the submission, I feel very sorry to reject this paper.

---

### Decision · Program_Chairs · 2020-07-31

**Decision:**

Reject

**Comment:**

It's been a hard but constructive discussion, but the reviews show from different aspects that this contribution will not be accepted for our workshop.
We encourage the authors to keep up their efforts in the field and act upon the suggestions made. We would love to see a submission from you next year. We cordially invite you dial in for the workshop itself to be part of our community and make contributions there. We are looking forward to hearing from you in the future.